# A Comparison of Gene Expression Profiles of Rat Tissues after Mild and Short-Term Calorie Restrictions

**DOI:** 10.3390/nu13072277

**Published:** 2021-06-30

**Authors:** Kenji Saito, Maiko Ito, Takuya Chiba, Huijuan Jia, Hisanori Kato

**Affiliations:** 1Health Nutrition, Department of Applied Biological Chemistry, Graduate School of Agricultural and Life Sciences, The University of Tokyo, 1-1-1 Yayoi, Bunkyo-ku, Tokyo 113-8657, Japan; skkj774@gmail.com (K.S.); mi@mai.co.jp (M.I.); ginajhj77@yahoo.co.jp (H.J.); 2Biomedical Gerontology Laboratory, Faculty of Human Sciences, Waseda University, 2-579-1 Mikajima, Tokorozawa, Saitama 359-1164, Japan; takuya@waseda.jp

**Keywords:** calorie restriction, gene expression profiles, nutrigenomics, rat

## Abstract

Many studies have shown the beneficial effects of calorie restriction (CR) on rodents’ aging; however, the molecular mechanism explaining these beneficial effects is still not fully understood. Previously, we conducted transcriptomic analysis on rat liver with short-term and mild-to-moderate CR to elucidate its early response to such diet. Here, we expanded transcriptome analysis to muscle, adipose tissue, intestine, and brain and compared the gene expression profiles of these multiple organs and of our previous dataset. Several altered gene expressions were found, some of which known to be related to CR. Notably, the commonly regulated genes by CR include nicotinamide phosphoribosyltransferase and heat shock protein 90, which are involved in declining the aging process and thus potential therapeutic targets for aging-related diseases. The data obtained here provide information on early response markers and key mediators of the CR-induced delay in aging as well as on age-associated pathological changes in mammals.

## 1. Introduction

Many studies have shown that calorie restriction (CR) has beneficial health effects on rodents and human [1,2]. Several molecular mechanisms, including the oxidative stress stimulation and nutrient-sensing pathways, have been proposed to explain the role of CR for improving health and extending life [3]. Recent advances in science and technology have allowed omics-based (e.g., transcriptomics, proteomics, metabolomics) deep investigation of the molecular mechanism of CR [4,5]. However, it is still unclear how CR delays the aging process and the surge of age-related diseases.

The liver, muscle, adipose tissue, intestine, and brain are main organs with pivotal roles in the metabolism and absorption of food and nutrients. These organs, which are known to communicate with each other [6,7], are significant for the regulation of homeostasis by nutrient sensing. In addition, all organs and tissues are affected by CR, a major protective factor for most age-related diseases in humans. Therefore, investigating the commonalities and differences among multiple organs in response to CR is an important insight that can help disclosing the molecular mechanisms underlying the effects of CR.

In general, any experiment related to CR is performed in a longevity and aging study that requires a long-term observation. Reported animal studies were conducted under relatively long-term and severe CR conditions such as 30% dietary restriction for several months, which is difficult to achieve in humans. However, transcriptomic alterations usually occurs much earlier than phenotypic alterations thereby enabling us to find early response markers of CR.

Previously, we examined the liver transcriptome of rats with short-term (one week and one month) and mild-to-moderate (5%, 10%, 20%, and 30%) CR [8]. In the present study we expanded transcriptome analysis to other target organs, such as muscle, adipose tissue, intestine, and brain, and compared the gene expression profiles of these organs among them and to our previous dataset to find early response markers and key mediators of the CR-induced beneficial effects related to age.

## 2. Materials and Methods

### 2.1. Animal Experiment

The methods used in the animal experiment have been described previously [8]. Male Wistar rats purchased at 5 weeks of age from Japan SLC (Tokyo, Japan) were kept individually and fed the American Institute of Nutrition-93G powdered diet ad libitum for 1 week or 1 month. The daily intakes of these two groups were recorded, and then 100%, 95%, 90%, 80%, and 70% of their daily intakes were provided to five groups (control, 5, 10, 20, and 30% CR, respectively) of five rats each for 1 week or 1 month. (Figure 1). The rats were anesthetized with diethyl ether on the last day of the experiment, after an overnight (16 h) fast and their white adipose tissue (visceral fat), skeletal muscle (gastrocnemius), brain (hypothalamus area), small intestine (collected only after 1 month), and liver (data reported previously) were collected. All procedures were done according to the Animal Usage Committee of the Faculty of Agriculture at the University of Tokyo’s regulations, and the committee’s consent was acquired (permission number 1818T0011).

### 2.2. Microarray Experiments

Total RNA was extracted from adipose tissue, the hypothalamus, muscle, and the intestine for the microarray experiment. As previously stated, total RNA was extracted from these tissues using the RNeasy mini kit (Qiagen, Valencia, CA, USA) [8]. The DNA microarray analysis was performed with the Affymetrix GeneChip according to the standard Affymetrix (Thermo Fisher Scientific, Santa Clara, CA, USA) protocols. For the adipose tissue analysis, a pool of complementary RNA was divided in half and used separately for the hybridization to the Affymetrix GeneChip Rat Expression Set 230 Array. A pool of complementary RNA was employed for hybridization to the Affymetrix GeneChip Rat Genome 230 2.0 Array for muscle and hypothalamic investigations. A pool of complementary RNA was used to hybridize to the Affymetrix GeneChip Rat Expression Set 230 Array for the small intestine analysis.

### 2.3. Differentially Expressed Gene Probes

As previously described [8], we used the default criteria of the Affymetrix GeneChip Operating Software MAS5.0, which was used as follows: ‘detection *p* value’—present, *p* < 0.04; marginal, 0.04 ≤ *p* < 0.06; and absent, *p* ≥ 0.06; and ‘change *p* value’—increase, *p* ≤ 0.0025; marginal increase, 0.0025 < *p* ≤ 0.003; decrease, *p* ≥ 0.998; marginal decrease, 0.997 ≤ *p* < 0.998; and no change, 0.003 < *p* < 0.997. An algorithm was used to generate signal log ratio, which is a quantitative estimate of the change in gene expression. We did not use a fold-change cut-off to get dynamic expression changes generated by mild-to-moderate CR. A conservative approach in the analysis with a combination of stringent filtering methods was used to reduce false positives. Probe sets that were ‘absent’ in at least one of each hybridization pair were excluded. Comparisons with a ‘no change’ and a ‘marginal increase’ and a ‘marginal decrease’ call were eliminated. For the adipose tissue analysis, we used duplicate GeneChips were used on each group sample, and the expression change was taken as informative if the change call of both chips was either ‘increase’ or ‘decrease’, and in the same direction. We used one GeneChip on each group sample for the muscle, hypothalamus, and small intestine studies, and the expression change was considered informative if the change call was either ‘increase’ or ‘decrease’. The probe sets that showed changes in the same direction across all levels of CR were regarded to be “CR responsive” genes. Microsoft Excel (Microsoft Corp., Redmond, WA, USA) was used to filter the data and identify probe sets that overlapped. The GEO site (http://www.ncbi.nlm.nih.gov/geo/, accessed on 27 June 2021) contains raw data set for all tissues (accession number GSE18297 and GSE176300). The change of the informative gene expression was further validated using randomly selected gene probes in the liver of rats, as previously described [8].

### 2.4. Gene Ontology Analysis

The functional annotation tool of the Database for Annotation, Visualization, and Integrated Discovery (DAVID) 6.8 was used to undertake gene ontology (GO) analysis on the 30% CR group, which received the most robust CR intervention [9]. This web-based functional annotation tool picks up enrichment in gene groups corresponding to biological functions or categories. For the analysis of CR responsive genes, all informative genes defined were used. For the comparison between each CR treatments and its respective controls, Gene Ontology: Biological Process categories (BP_Direct) were significantly over-represented, as determined by Fisher’s exact test (Adjusted *p*-value < 1 × 10^−4^ by the Benjamini and Hochberg method). We also re-analyzed the gene expression data previously obtained for the liver using this version of DAVID.

## 3. Results

### 3.1. The Number of CR Responsive Genes

The number of gene probes that were changed by CR was different among the examined tissues (Table 1). There are fewer gene probes in the 5% and 10% CR groups than in the 20% and 30% CR groups for the one week (1 w) liver and adipose tissues and one month (1 m) adipose tissue and intestine experiments. Additional details regarding the number of gene probes that were altered are shown in Appendix A.

### 3.2. CR Responsive Genes for Each Tissue

The top five gene probes that were consistently up- or down-regulated across all CR levels (defined as CR responsive genes) for each tissue are displayed in Table 2 and Table 3. The values for the change in expression of each gene are shown on a log scale and are sorted according to the values found for the 30% CR groups after one month of experiment, as this was the most robust intervention in the present study.

Our previous study demonstrated a significant up-regulation of metallothionein genes (namely *Mt2a* and *Mt1a*), involved in metal detoxification and oxidative stress, and down-regulation of fatty acid synthase (*FAS*) genes, which code for key enzymes tangled in fatty acid synthesis in rat liver of rats with mild-to-moderate CR [8]. These major findings were confirmed here by the re-analysis of the GO data obtained for the 30% CR group using DAVID. Both the up-regulation of the ‘oxidation-reduction process [GO:0055114] (*p* = 8.65 × 10^−13^)’ and ‘fatty acid beta-oxidation [GO:0006635] (*p* = 1.39 × 10^−6^)’ categories (Appendix A) and the down-regulation of ‘lipid metabolic process [GO:0006629] (*p* = 4.03 × 10^−5^)’ were demonstrated (Appendix A).

Table 2 and Table 3 demonstrate that 45 and 14 gene probes were up- and down-regulated in adipose tissue, respectively, as compared to the control group. The D site of the albumin promoter (albumin D-box) binding protein (*Dbp*) gene, which is involved in circadian rhythm regulation, was shown to be the most up-regulated. The steroidogenic acute regulatory protein (*Star*) gene, which codes for the rate-limiting enzyme in the production of steroid hormones from cholesterol [10], was also significantly up-regulated. Most significantly down-regulated genes included the secretory leukocyte peptidase inhibitor (*Slpi*) and the adipose-derived inflammatory factor that is reported to be increased with obesity [11]. These genes code for important lipid-metabolizing enzymes. Correspondingly, GO analysis identified the up-regulation of the ‘tricarboxylic acid cycle’ [GO:0006099] (*p* = 1.63 × 10^−13^) and ‘fatty acid beta-oxidation‘ [GO:0006635] (*p* = 1.1 × 10^−11^) in the top-ranked categories (Appendix A).

Fifty genes were up-regulated, and 51 genes were down-regulated in muscle. Flavin containing monooxygenase 1 (*FMO1*), a novel regulator of energy balance that promotes metabolic efficiency, was one of the top-ranked genes that were up-regulated. The G0/G1 switch 2 (*G0s2*), which controls lipid metabolism in muscle [12], was significantly down-regulated. GO analysis indicated that the ‘response to hypoxia‘ [GO:0001666] (*p* = 1.29 × 10^−4^), ‘muscle contraction‘ [GO:0006936] (*p* = 3.5 × 10^−4^), and ‘actin cytoskeleton organization‘ [GO:0030036] (*p* = 5.90 × 10^−4^) were significantly over-represented in the down-regulated category (Appendix A).

The number of altered gene probes at the 1 m-hypothalamus sample was small; hence, only two known genes were identified as down-regulated and as CR responsive genes in this tissue. One of these genes, the nuclear receptor subfamily 4, group A, member 3 (*Nr4a3*), may promote food intake as well as body weight gain via its actions in the brain [13]. Another gene, the early growth response 1 (*Egr1*), is known as a neurogenic transcription factor and it is associated with appetite signaling [14].

Forty-seven genes were up-regulated and 23 genes were down-regulated in the small intestine. The top five up-regulated genes included sulfotransferase family 1A member 1 (*Sult1a1*), which is highly expressed in the small intestine [15] and is important in xenobiotic metabolism. The down-regulated genes included the gut peptide neuromedin U (*Nmu*) that has been reported to decrease food intake and body weight [16]. Interestingly, ‘aging‘ [GO:0007568] (*p* = 3.93 × 10^−4^) category was up-regulated in the intestine (Appendix A).

### 3.3. Up-Regulated Genes across All Tissues Studied

As we could not identify all CR responsive genes across all tissues, the genes that exhibited the greatest changes across multiple tissues were screened. The top ten up-regulated genes, including at least 20 out of 36 CR conditions across multiple organs, are listed in Table 4. The most commonly up-regulated gene was *Nr1d2*, an orphan nuclear receptor known as a circadian regulator. This gene was up-regulated in almost all CR conditions except in the 1w-hypothalamus. Furthermore, the aldo-keto reductase family 1, member C14 (*Akr1c14*) and the nicotinamide phosphoribosyltransferase (*Nampt*) genes were up-regulated within 21 CR conditions. Other up-regulated genes across 20 CR conditions included the glutamate-ammonia ligase (*Glul*), sulfotransferase family 1A member 1 (*Sult1a1*), CD36 molecule (thrombospondin receptor) (*Cd36*), flavin containing monooxygenase 1 (*Fmo1*), epoxide hydrolase 1, microsomal (xenobiotic) (*Ephx1*), and *Dbp*.

The expression signatures differed among tissues. For example, no up-regulated genes were observed in the brain, except for *Nr1d2* and *Sult1a1*. Genes *Sult1a1*, *Ephx1*, and *Dbp* were not found in the liver, and no changes in *Fmo1* expressions were observed in the brain and intestine.

A complete list of the up-regulated genes across the multiple tissues is displayed in Appendix A.

### 3.4. Down-Regulated Genes across All Tissues Studied

Genes down-regulated in response to CR in at least 19 of 36 CR conditions among the multiple tissues are also listed in Table 4. The most commonly down-regulated gene was CKLF-like MARVEL transmembrane domain containing 6 (*Cmtm6*), which is tangled in immune response and inflammatory activities. Cytochrome P450, family 51 (*Cyp51*), known to be involved in cholesterol biosynthesis, was ranked second. The heat shock protein 90, alpha (cytosolic), class A member 1 (*Hsp90aa1*) was ranked the third most commonly down-regulated gene, although no change was detected in the liver tissue. Genes coding for structural proteins such as actin, gamma 1 (*Actg1*), tubulin, beta 4B class IVb (*Tubb4b*), and tubulin, beta 2A class IIa (*Tubb2a*) were down-regulated as expected. Additionally, fatty acid desaturase 1 (*Fads1*), tropomyosin 4 (*Tpm4*), and sphingosine-1-phosphate receptor 1 (*S1pr1*) were down-regulated.

No changes in *Cmtm6* and *Actg1* expressions were detected in the intestine. No changes in the expressions of *Tubb4b*, *Tpm4*, *S1pr1*, and *Tubb2a* were detected in the hypothalamus and intestine.

### 3.5. Comparison with a Meta-Analysis of the CR Effect

Two meta-analyses have been conducted for 33 and 61 CR studies [17,18], as summarized in Appendix A. We compared our 36 CR responsive genes found in each tissue with the findings reported for these CR meta-analyses datasets. The overlapped genes are shown in Appendix A. Eighteen of our 36 CR responsive genes overlapped with those of the previous meta-analyses, and all gene expressions except that of *Dbp* were changed in the same direction as our data.

The 18 most commonly regulated genes across all tissues were also compared using the CR meta-analyses. Seven of these 18 genes overlapped with the CR meta-analyses data (Table 5). More specifically, genes *Nampt*, *Glul*, *Sult1a1*, and *Fmo1* were up-regulated, while *Cmtm6*, *Cyp51*, *Hsp90aa1*, *Actg1*, and *Tubb2a* were down-regulated in both our study and in the meta-analyses. The *Sult1a1* and *Actg1* genes overlapped in both meta-analyses datasets. Overall, nearly half of the genes found in our study were overlapped with those in the previous CR meta-analyses.

## 4. Discussion

The gene expression profiles of numerous tissues of young growing rats with mild-to-moderate and short-term CR were studied in this work. First, we presented the transcriptomic characteristics induced by CR in each tissue (Table 2 and Table 3). Notably, by comparing the CR responsive genes in each tissue found in our study with the dataset resulting from two previous meta-analyses, which contained severe CR conditions in multiple organs and tissues, nearly half of the genes were overlapped in the same direction (Appendix A). The remaining genes did not overlap or conflicted (only one gene) with those in the meta-analyses dataset, and this discrepancy may explain the differential responses to CR in the setting of each study (e.g., duration and strength of CR, dietary regimes, gender, species and developmental stage of animals used).

Second, we looked for a common element that could cause the favorable effect of CR by looking at the commonality of genes reacting to CR across diverse tissues. The top 18 genes that were consistently altered across the five examined tissues were therefore further evaluated (Table 4). Among them, seven genes (*Nampt*, *Glul*, *Sult1a1*, *Fmo1*, *Cmtm6*, *Hsp90aa1*, and *Actg1*) were overlapped with the CR meta-analyses dataset in the same direction (Table 5). These genes might be essential for CR beneficial effects and thus might be used as sensitive biomarkers of CR, as they responded to relatively mild CR conditions and in short-term in multiple tissues.

In a wide range of species, including rats and primates, it is commonly acknowledged that CR extends lifespan. We found several aging-related genes that changed across the multiple tissues examined and are represented by *Nampt*. This gene codes for the major rate-limiting enzyme in NAD+ production, which has been shown to decline with age in multiple tissues such as the liver, adipose, and brain [19]. The pathogenesis of age-related metabolic disorders is aided by the lowering of *Nampt* and NAD+ levels in many tissues [20,21]. A study performed using an eight-week CR also demonstrated the up-regulation of *Nampt* mRNA in rat organs [22], consistent with our data.

Another key biological process and target underlying aging is cellular senescence. We found the down-regulation of *Hsp90aa1*, a member of the Hsp90 superfamily. This gene codes for an abundant protein that functions as a molecular chaperone [23]. The inhibitors of Hsp90 have been shown recently to be a novel class of senolytics [24]. Multi-tissue dysregulation of Hsp90 members, as seen in our study, may explain the anti-aging effect of CR through the clearance of senescent cells. Moreover, the up-regulation of *Glul*, known to code for the glutamine synthetase that catalyzes the *de novo* synthesis of glutamine from glutamate and ammonia, was observed. *Glul* is ubiquitously expressed and particularly highly expressed in the muscle, liver, and brain [25]. Recently, Johomura et. al. showed that activation of glutaminolysis induced the production of ammonia, which neutralized the lower pH to improve the survival of the senescent cells [26]. Therefore, the up-regulation of *Glul* induced by CR may inversely repress this process, thus promoting the death of senescent cells as well as a decline of aging effects.

One of the most prominent health benefits of CR is the prevention of malignancies. We found that cancer-related genes such as *Cmtm6* were down-regulated in response to CR. CMTM6 is a ubiquitously expressed protein that is known to be a critical regulator of PD-L1 [27,28], a target of immune checkpoint inhibitor therapy [29]. CMTM6 depletion in multiple tissues may decrease PD-L1 expression and cancer incidence through a CR-related mechanism. Moreover, *S1pr1*, a novel promising target in cancer therapy [30], was broadly down-regulated across the five tissues examined in our study. Thus, CR-induced metabolic changes may not only reduce the incidence of cancer, but also increase the efficacy of cancer therapies as proposed previously [31].

According to a current hypothesis of aging, aging is caused by a loss in detoxifying capacity, and CR affects the expressions of genes involved in xenobiotic metabolism [32]. In our study, the xenobiotic-related gene *Fmo1* was up-regulated and overlapped with the meta-analyses dataset. FMOs are enzymes originally implicated in the oxidation of xenobiotics but have been recently implicated to promote longevity and health span [33,34]. EPHX1 is an enzyme that aids in the detoxification of cigarette-related chemicals [35], which is a significant risk factor for the development of certain cancers [36]. SULT1A1 is a phase II xenobiotic metabolizing enzyme that is extensively expressed in the liver and facilitates carcinogen sulfonation [37]. It is therefore no surprise that these detoxication enzymes are transcriptionally activated in response to CR in multiple tissues, and that they may contribute to the extension of the human lifespan.

Even minor calorie restriction, such as a 15% dietary restriction for 16 weeks, has been shown to enhance lipid metabolism [38]. We also observed a marked decrease of white adipose tissue weight under a 20% CR diet for one week [8]. However, many genes related to lipid metabolism were identified that did not overlap the meta-analyses dataset. *Cd36*, which is well known to be tangled in the regulating of lipogenesis in human adipose tissue [39] was up-regulated. CYP51, known to have a role in cholesterol biosynthesis in mammalian cells [40], and FADS1, a rate-limiting enzyme that generates long-chain polyunsaturated fatty acids [41], were down-regulated. As expected, the expression of these genes was mostly changed toward improving lipid metabolism in target tissues. Moreover, *Nr1d2* (also referred to Rev-erbbeta), first discovered to be a gene regulator involved in circadian rhythm and fat accumulation [42], was most ubiquitously up-regulated in this study. It also controls lipid and energy homeostasis in skeletal muscle [43]. Interestingly, circadian clock gene was recently proposed to mediate the beneficial effect of CR and that may contribute to longevity [44].

Our present findings demonstrate that the expressions of key genes involved in the CR-induced beneficial effects were regulated even by very mild and short-term interventional CR conditions, such as 5%–10% CR for one week, which are applicable in humans. These expression profiles can also be used as reference data for removing the transcripts responding to the reduction of food intake often observed in in vivo nutrition research. Indeed, CR profiles have been successfully applied to identify sensitive transcriptomic biomarkers of selenium status [45]. In summary, the approach and data obtained in the present study will not only help providing insight on novel mechanisms associated with CR-induced health benefits but may also identify targets for functional and safety assessment of food and nutrients.

One limitation of our investigation is the absence of biological replicates, as we used pooled samples for DNA microarray analysis. However, we applied conservative criteria to screen the genes in the data filtering process and the changes in expression were partly validated by qPCR using randomly selected samples in a previous study [8]. Another limitation is that we used relatively young rats for the CR study since one of the purposes of this investigation was to obtain the reference gene expression data for in vivo nutrition research where the young growing rats are often used.

## 5. Conclusions

Our findings give essential knowledge on the molecular mechanisms underlying CR’s positive consequences. The present study also provides a way to identify dietary or therapeutic targets for aging-related diseases. Further studies are required to determine if the genes found in this study are involved in the essential mechanism of CR-induced beneficial effects in different species, including human.

## Figures and Tables

**Figure 1 nutrients-13-02277-f001:**
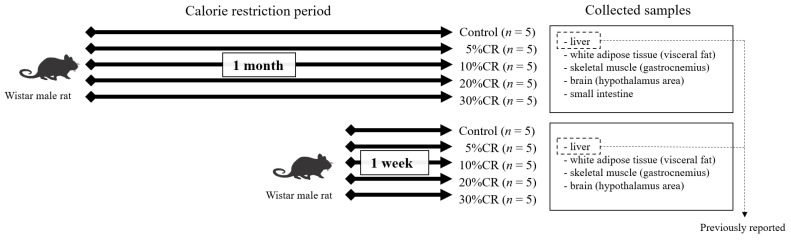
Overview of the animal experiment.

**Table 1 nutrients-13-02277-t001:** Number of genes altered by short-term and mild-to-moderate CR.

Organs	Duration	5%	10%	20%	30%	Overlap ^†^
liver	1 week	183	588	902	734	90
	1 month	495	454	572	570	
adipose	1 week	387	981	1084	1674	59
	1 month	304	1232	1624	1255	
muscle	1 week	684	663	1229	1517	101
	1 month	1041	1097	1074	1422	
brain (hypothalamus)	1 week	1346	1413	1429	1345	3
	1 month	76	58	119	148	
intestine	-	-	-	-	-	70
	1 month	370	503	530	558	

^†^ The number of gene probes overlapped in the same direction throughout all the CR groups. The values were determined according to the criteria described in Materials and methods. 5%: 5% calorie restriction. 10%: 10% calorie restriction. 20%: 20% calorie restriction. 30%: 30% calorie restriction.

**Table 2 nutrients-13-02277-t002:** Top five gene probes consistently up-regulated across all CR levels for each tissue.

		1 Week				1 Month				Gene Title	Gene Symbol
	Probe ID	5%	10%	20%	30%	5%	10%	20%	30%		
liver (40 gene probes)	1388271_at	2.2	3.5	4.5	4.5	2.3	2.9	3.4	4.0	metallothionein 2A	Mt2A
	1371237_a_at	2.4	3.6	4.1	4.0	2.9	3.4	3.5	3.9	metallothionein 1a///transthyretin	Mt1a///Ttr
	1387336_at	1.0	2.4	3.7	3.7	2.0	2.6	3.2	3.5	N-acetyltransferase 8	Nat8
	1387156_at	3.9	4.4	4.4	4.1	3.9	4.1	3.8	3.3	hydroxysteroid (17-beta) dehydrogenase 2	Hsd17b2
	1368213_at	0.8	1.5	2.5	3.1	1.5	2.0	2.5	2.8	P450 (cytochrome) oxidoreductase	Por
adipose (45 gene probes)	1387874_at	2.5	2.7	2.9	2.9	2.8	2.9	3.0	2.5	D site of albumin promoter (albumin D-box) binding protein	Dbp
	1388039_a_at	1.0	1.7	2.0	2.4	1.6	2.3	2.3	2.4	gamma-aminobutyric acid (GABA) B receptor 1	Gabbr1
	1368406_at	1.2	1.6	2.3	2.4	1.4	1.7	2.2	2.3	steroidogenic acute regulatory protein	Star
	1372536_at	0.8	1.2	1.9	2.3	0.9	1.4	1.9	2.2	aarF domain containing kinase 3	Adck3
	1387174_a_at	1.0	1.5	1.9	2.0	1.5	1.8	2.2	2.2	steroidogenic acute regulatory protein	Star
muscle (50 gene probes)	1378927_at	0.8	1.0	1.6	1.6	1.4	1.5	1.8	1.8	-	-
	1387053_at	0.6	1.0	0.9	1.5	0.9	0.9	1.1	1.6	flavin containing monooxygenase 1	Fmo1
	1368971_a_at	0.7	0.8	1.0	1.1	1.2	1.1	1.3	1.5	synaptojanin 2	Synj2
	1369150_at	0.3	0.9	0.6	1.2	0.7	0.6	0.8	1.4	pyruvate dehydrogenase kinase, isozyme 4	Pdk4
	1370019_at	0.7	0.9	1.2	1.4	1.0	1.1	1.2	1.4	sulfotransferase family 1A member 1	Sult1a1
brain	not identifed										
intestine (47 gene probes)	1370019_at					1.2	1.3	1.3	1.8	sulfotransferase family 1A member 1	Sult1a1
	1371076_at					1.3	1.1	2.0	1.8	cytochrome P450, family 2, subfamily b, polypeptide 1///cytochrome P450, family 2, subfamily b, polypeptide 2	Cyp2b1///Cyp2b2
	1368303_at					1.3	1.3	1.6	1.4	period circadian clock 2	Per2
	1367774_at					0.5	1.0	0.7	1.3	glutathione S-transferase alpha 1///glutathione S-transferase alpha-3-like	Gsta1///LOC102550391
	1369455_at					1.3	0.9	1.9	1.2	ATP-binding cassette, subfamily G (WHITE), member 5	Abcg5

Values are shown as log ratio vs control.

**Table 3 nutrients-13-02277-t003:** Top five gene probes consistently down-regulate across all CR levels for each tissue.

		1 Week				1 Month				Gene Title	Gene Symbol
	Probe ID	5%	10%	20%	30%	5%	10%	20%	30%		
liver (50 gene probes)	1367707_at	−1.55	−2.45	−4.3	−4.4	−1.7	−2.85	−3.6	−4.4	fatty acid synthase	Fasn
	1367708_a_at	−1.2	−2.15	−4	−4.3	−1.6	−2.75	−3.5	−3.9	fatty acid synthase	Fasn
	1373718_at	−1.6	−2.9	−3.75	−4.05	−2.55	−3.15	−3.45	−3.4	tubulin, beta 2A class IIa	Tubb2a
	1370870_at	−1.5	−1.9	−2.95	−2.95	−1.75	−2.25	−2.55	−2.85	malic enzyme 1, NADP(+)-dependent, cytosolic	Me1
	1367854_at	−1.1	−1.85	−2.55	−2.8	−1.4	−2	−2.3	−2.55	ATP citrate lyase	Acly
adipose (14 gene probes)	1367998_at	−1.15	−1.4	−1.35	−2.85	−0.7	−2.5	−2.45	−1.95	secretory leukocyte peptidase inhibitor	Slpi
	1368294_at	−1.1	−0.5	−1.4	−2.3	−0.9	−1.45	−1.8	−1.8	deoxyribonuclease 1-like 3	Dnase1l3
	1389006_at	−0.75	−0.7	−1.2	−2	−0.7	−1.2	−1.6	−1.6	macrophage expressed 1	Mpeg1
	1368189_at	−0.9	−0.65	−1.45	−1.7	−0.9	−0.85	−1.3	−1.4	7-dehydrocholesterol reductase	Dhcr7
	1373718_at	−0.45	−0.8	−0.8	−0.95	−0.75	−0.95	−0.8	−1.15	tubulin, beta 2A class IIa	Tubb2a
muscle (51 gene probes)	1388395_at	−0.7	−1.7	−2.5	−2.9	−2.5	−3.5	−3	−2.2	G0/G1switch 2	G0s2
	1378423_at	−0.3	−0.9	−1.3	−1.6	−1.8	−1.7	−1.8	−2.1	nicotinamide riboside kinase 2	Nmrk2
	1379416_at	−1.1	−1.4	−2	−2.2	−1.8	−1.5	−2.5	−2	autism susceptibility candidate 2-like	LOC100362819
	1378586_at	−2.5	−2.2	−2.2	−2.2	−2	−1.7	−0.6	−1.9	cytokine inducible SH2-containing protein	Cish
	1374204_at	−1.4	−1.1	−1.8	−1.9	−2.5	−2.6	−2.5	−1.4	WD repeat and SOCS box-containing 1	Wsb1
brain (3 gene probes)	1375043_at	−2.3	−2.4	−2	−2.1	−1.3	−1.4	−1.2	−1.4	-	---
	1369067_at	−1	−1	−0.9	−1.1	−0.9	−1.2	−1.2	−1.1	nuclear receptor subfamily 4, group A, member 3	Nr4a3
	1368321_at	−0.7	−0.7	−0.4	−0.7	−1.2	−1.4	−1.2	−0.9	early growth response 1	Egr1
intestine (23 gene probes)	1369717_at					−1	−1.1	−1.1	−1.7	neuromedin U	Nmu
	1387758_at					−0.5	−1	−0.2	−1.7	alkaline phosphatase 3, intestine, not Mn requiring	Akp3
	1378658_at					−0.9	−1.6	−1	−1.4	chloride channel accessory 4	Clca4
	1368247_at					−1.4	−1.1	−1.1	−1.2	heat shock 70kD protein 1A///heat shock 70kD protein 1B (mapped)	Hspa1a///Hspa1b
	1389986_at					−0.7	−0.5	−1.1	−1.1	-	---

Values are shown as log ratio vs control.

**Table 4 nutrients-13-02277-t004:** Top-ranked genes commonly regulated by CR across all tissues studied.

**Probe ID**	**Gene** **Symbol**	**Gene Title**	**# of Overlapped** **Gene Probes**	**Liver**								**Adipose**								**Muscle**								**Brain(hypothalamus)**								**Intestine**			
				**1 Week**				**1 Month**				**1 Week**				**1 Month**				**1 Week**				**1 Month**				**1 Week**				**1 Month**				**1 Month**			
				**5%**	**10%**	**20%**	**30%**	**5%**	**10%**	**20%**	**30%**	**5%**	**10%**	**20%**	**30%**	**5%**	**10%**	**20%**	**30%**	**5%**	**10%**	**20%**	**30%**	**5%**	**10%**	**20%**	**30%**	**5%**	**10%**	**20%**	**30%**	**5%**	**10%**	**20%**	**30%**	**5%**	**10%**	**20%**	**30%**
1390430_at	Nr1d2	nuclear receptor subfamily 1, group D, member 2	29	↑	↑	↑	↑	↑	↑	↑	↑	↑	↑	↑	↑	↑	↑	↑		↑	↑	↑	↑	↑	↑	↑	↑					↑	↑			↑	↑	↑	↑
1370708_a_at	Akr1c14	aldo-keto reductase family 1, member C14	21		↑	↑	↑	↑	↑	↑	↑	↑	↑	↑	↑		↑	↑	↑	↑	↑				↑		↑									↑		↑	↑
1389014_at	Nampt	nicotinamide phosphoribosyltransferase	21		↑	↑	↑	↑	↑	↑	↑	↑	↑	↑	↑	↑	↑	↑	↑			↑	↑		↑											↑	↑		↑
1367633_at	Glul	glutamate-ammonia ligase	20		↑	↑	↑	↑				↑	↑	↑	↑	↑	↑	↑	↑		↑	↑	↑	↑	↑	↑	↑										↑		
1370019_at	Sult1a1	sulfotransferase family 1A member 1	20										↑	↑	↑			↑	↑	↑	↑	↑	↑	↑	↑	↑	↑					↑	↑		↑	↑	↑	↑	↑
1386870_at	Glul	glutamate-ammonia ligase	20		↑	↑	↑	↑			↑	↑	↑	↑	↑	↑	↑	↑	↑			↑	↑	↑	↑	↑	↑										↑		
1386901_at	Cd36	CD36 molecule (thrombospondin receptor)	20		↑	↑	↑					↑	↑	↑	↑			↑		↑	↑	↑	↑	↑	↑	↑	↑									↑	↑	↑	↑
1387053_at	Fmo1	flavin containing monooxygenase 1	20	↑	↑	↑	↑	↑	↑	↑	↑			↑	↑			↑	↑	↑	↑	↑	↑	↑	↑	↑	↑												
1387669_a_at	Ephx1	epoxide hydrolase 1, microsomal (xenobiotic)	20									↑	↑	↑	↑	↑	↑	↑	↑	↑	↑	↑	↑	↑	↑	↑	↑									↑	↑	↑	↑
1387874_at	Dbp	D site of albumin promoter (albumin D-box) binding protein	20									↑	↑	↑	↑	↑	↑	↑	↑	↑	↑	↑	↑	↑	↑	↑	↑									↑	↑	↑	↑
**Probe ID**	**Gene** **Symbol**	**Gene Title**	**# of Overlapped** **Gene Probes**	**Liver**								**Adipose**								**Muscle**								**Brain(hypothalamus)**								**Intestine**			
				**1 Week**				**1 Month**				**1 weEk**				**1 Month**				**1 Week**				**1 Month**				**1 Week**				**1 Month**				**1 Month**			
				**5%**	**10%**	**20%**	**30%**	**5%**	**10%**	**20%**	**30%**	**5%**	**10%**	**20%**	**30%**	**5%**	**10%**	**20%**	**30%**	**5%**	**10%**	**20%**	**30%**	**5%**	**10%**	**20%**	**30%**	**5%**	**10%**	**20%**	**30%**	**5%**	**10%**	**20%**	**30%**	**5%**	**10%**	**20%**	**30%**
1372056_at	Cmtm6	CKLF-like MARVEL transmembrane domain containing 6	25	↓	↓	↓	↓	↓	↓	↓	↓		↓	↓	↓	↓	↓	↓	↓	↓	↓	↓	↓	↓	↓	↓	↓	↓		↓									
1367979_s_at	Cyp51	cytochrome P450, family 51	23	↓	↓	↓	↓	↓	↓	↓	↓						↓	↓	↓	↓	↓	↓	↓	↓		↓	↓	↓	↓	↓	↓						↓		
1388850_at	Hsp90aa1	heat shock protein 90, alpha (cytosolic), class A member 1	21										↓	↓	↓	↓	↓	↓	↓	↓	↓	↓	↓	↓	↓	↓	↓	↓	↓	↓	↓						↓	↓	
1371327_a_at	Actg1	actin, gamma 1	20	↓	↓	↓	↓	↓	↓	↓	↓			↓	↓		↓	↓	↓		↓	↓	↓	↓	↓	↓	↓												
1367857_at	Fads1	fatty acid desaturase 1	19			↓	↓				↓		↓	↓	↓	↓	↓	↓	↓	↓	↓	↓	↓	↓	↓	↓	↓											↓	
1371390_at	Tubb4b	tubulin, beta 4B class IVb	19	↓	↓	↓	↓	↓	↓	↓	↓			↓	↓	↓	↓	↓	↓			↓		↓	↓	↓	↓												
1371653_at	Tpm4	tropomyosin 4	19			↓	↓	↓	↓					↓	↓		↓	↓	↓			↓	↓	↓	↓	↓	↓	↓	↓	↓	↓								
1371840_at	S1pr1	sphingosine-1-phosphate receptor 1	19		↓	↓	↓	↓	↓	↓			↓		↓		↓	↓	↓					↓	↓	↓	↓	↓	↓	↓	↓								
1372727_at	-	-	19	↓	↓	↓	↓	↓	↓	↓	↓	↓	↓				↓			↓	↓	↓	↓	↓	↓	↓	↓												
1373718_at	Tubb2a	tubulin, beta 2A class IIa	19	↓	↓	↓	↓	↓	↓	↓	↓	↓	↓	↓	↓	↓	↓	↓	↓					↓	↓		↓												

Red arrows represent the genes up-regulated and blue arrows represent the genes down-regulated by mild CR. Complete list is available at Appendix A.

**Table 5 nutrients-13-02277-t005:** The overlap of top-ranked genes commonly regulated by CR across all tissues with previous meta-analyses data.

Gene Symbol	Function	Our Study	Ref #1	Ref #2
Nampt	NAD metabolism	up		up
Glul	glutamate metabolism	up		up
Sult1a1	xenobiotic metabolism	up	up	up
Fmo1	xenobiotic metabolism	up		up
Dbp	circadian rhythm	up	up	down
Cmtm6	immune system	down		down
Hsp90aa1	chaperone protein	down		down
Actg1	structural protein	down	down	down

## Data Availability

The microarray data have been deposited in the GEO database (https://www.ncbi.nlm.nih.gov/geo/, accessed on 27 June 2021) under accession number GSE176300.

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
