# Peer review of "A Comparison of Gene Expression Profiles of Rat Tissues after Mild and Short-Term Calorie Restrictions"

_nutrients, 2021, doi:10.3390/nu13072277_

Round 1

Reviewer 1 Report

Dear Authors,

The manuscript (nutrients-1225638) presented for review is very interesting. I recommend the article for publication after technical correction, because the quality of Tables 2, 3, and 4 is not good. The manuscript is very well written and contains the important and current topic of nutrigenomics.

The experiment conducted on rats delivered much valuable information concerning the molecular mechanism of the beneficial effects of calorie restriction. It is very important for understanding the mechanism of lost weight and its positive effects in mammals organisms.

Applause for the authors who in a simple way explained that experiment and results achieved.

Reviewer

Author Response

We wish to express our appreciation to the Reviewer for his or her insightful comments, which have helped us significantly improve the paper.

Comment 1: I recommend the article for publication after technical correction, because the quality of Tables 2, 3, and 4 is not good.

Response:Thank you very much for the comment. As suggested by the reviewer, we have improved the quality of Tables 2, 3, and 4.

Reviewer 2 Report

Please, see the attachement.

Author Response

We wish to express our appreciation to the Reviewer for his or her insightful comments, which have helped us significantly improve the paper.

Comment 1:   I how many animals in each group was taking part in experiments? - authors noted in study limitations that, they pooled material for analysis so how largewere the groups?

Response: We thank the reviewer for pointing this out and agree that this should be included in the text. To address this, a sentence was added in the method section (see page 2 line 58).

Comment 2:  7 weeks of age is a moment of sexual maturity assessment, so I think that animals used in the study should be described as young adult or even juvenile animals and an age should be also discussed in context of experiments over CR;

Response: We thank for pointing this out. As the reviewer suggested, it is important and should be discussed in the manuscript. We have added a sentence (see page 7 line 233) and limitation section (see page 9 line 319) as follows;

"Another limitation is that we used relatively young rats for the CR study since one of the purposes of this investigation was to obtain the reference gene expression data for in vivo nutrition research where the young growing rats are often used. "

Comment 3:Increase the sizes of all tables in main text: they are really poorly visible after printing the manuscript;

Response: We thank the reviewer for pointing this out. As suggested by the reviewer, we have improved the quality of Tables 2, 3, and 4.

Comment 4: Table 5: add a column explaining in which pathway a given gene is involved - it’s going to be more understandable for those who are not perform strictly genetic or biochemical research but are interested in CR effects;

Response: We think this is an excellent suggestion. As suggested by the reviewer, we have added the function column in the Table 5 to be more understandable.

Comment 5: Nr1d2 was a gene found up-regulated in all tissues but it was not discussed in discussion section : authors focused only on genes which were previously showed in other publications (showed in table 5), this could be interesting how gene from circadian regulation pathway influences longevity; add a short fragment to discuss this issue;

Response:  We thank for this suggestion. We added the short discussion in the manuscript as highlighted in yellow (see page 8 line 303) as follows;

" Interestingly, circadian clock gene was recently proposed to mediate the beneficial effect of CR and that may contribute to longevity [44]."

Comment 6:Some genes were not overlapping in the analysis with previous - authors should discuss why: different feeding regimes age gender or even species of animal model used for CR study?

Response: We thank for this suggestion. We have added sentences in the manuscript as highlighted in yellow (see page 7 line 241) as follows;

"(eg. duration and strength of CR, dietary regimes, gender, species and developmental stage of animals used)."

Comment 7: Reference 25 is not correctly cited: please check this is a review article not a original study and it’s not study of Johomura et al. but Zhang J.

Response: We thank for pointing this out. As reviewer commented, it's inappropriately cited. The citated reference was replaced by appropriate one.